# Adherences to oral nutritional supplementation among hospital outpatients: An online cross-sectional survey in Japan

**Naoki Hashizume**[1]*, **Yoshiaki Tanaka**[1,2], **Suguru Fukahori**[1], **Shinji Ishii**[1], **Nobuyuki Saikusa**[1], **Yoshinori Koga**[1], **Naruki Higashidate**[1], **Daisuke Masui**[1], **Saki Sakamoto**[1], **Minoru Yagi**[1]

1 Department of Pediatric Surgery, Kurume University School of Medicine, Kurume, Fukuoka Japan,
2 Division of Medical Safety Management, Kurume University Hospital, Kurume, Fukuoka, Japan

* n_hashidume@med.kurume-u.ac.jp

**Data Availability Statement:** All relevant data are within the manuscript.

**Funding:** The authors received no specific funding for this work.

## Abstract

Oral nutritional supplements (ONS) are multi-nutrient products used to increase the energy and nutrient intakes of patients. The aim of this study was to examine whether or not the adherence of patients varies according to their receiving prescription or over-the-counter ONS. Data were obtained from an online cross-sectional survey conducted with patients in Japan. A total of 107 patients who matched the inclusion criteria for the prescription ONS group and 148 who matched the criteria for the over-the-counter ONS group were further analyzed. In the prescription and over-the-counter ONS groups, the main medical reason for ONS consumption were "malnutrition" (48 patients [44.9%] vs. 63 patients [42.6%] p = 0.798], "frailty" (29 patients [27.1%] vs. 36 patients [24.3%] p = 0.663) and "aging" (25 patients [23.4%] vs. 30 patients [20.3%] p = 0.644). The proportion of "No particular disease" for prescription ONS consumption was significantly lower than that for over-the-counter ONS (6 patients [5.6%] vs. 24 patients [16.2%] p = 0.001). The body mass index of the prescription ONS group was significantly higher than that of the over-the-counter ONS group (21.1±4.38 kg/m$^2$ vs. 19.9±3.75 kg/m$^2$, p = 0.0161). In the prescription ONS group, all patients were given medical advice by doctors or registered dietitians. In contrast, in the over-the-counter ONS group, only 46 patients (31.1%) were given advice by doctors or registered dietitians (p<0.001). In the prescription ONS group, ONS was taken significantly more times and for a longer duration than in the over-the-counter ONS group (p<0.0001). However, among patients given advice by doctors or registered dietitians, there were no significant differences between the groups. Greater support by the medical team is still needed in order to maximize adherence to supplementation, especially concerning the calories, timing and period, so that benefits can be achieved and sustained.

## Introduction

Oral nutritional supplements (ONS) are multi-nutrient products (ready-made liquid, pudding or powder to be mixed with fluids) used to increase the energy and nutrient intakes of patients,

**Competing interests:** The authors have declared that no competing interests exist.

especially those with malnutrition and at nutritional risk [1]. The European Society for Clinical Nutrition and Metabolism (ESPEN) introduced the concept of ONS to the ESPEN guidelines on enteral nutrition [2]. ONS are defined as supplementary oral products consumed along with the normal diet for special medical purposes.

In meta-analyses, ONS have been shown to be clinically effective in some patient groups [3–7], such as malnourished geriatric patients [3,4], whereas a Cochrane review on disease-related malnutrition found no major differences in morbidity or mortality between patients receiving dietary advice and those prescription ONS [8]. The ESPEN guidelines strongly recommend that malnourished polymorbid medical inpatients or those at high risk of malnutrition who can safely reach their nutritional requirements orally be considered for ONS high in energy and protein in order to improve their nutritional status and quality of life. It further recommends that nutrient-specific ONS be administered to malnourished polymorbid medical inpatients or those at high risk of malnutrition when they may maintain muscle mass, reduce mortality or improve their quality of life with such a prescription, and that ONS be considered for polymorbid medical inpatients who are malnourished or at high risk of malnutrition and can safely reach their nutritional requirements orally as a cost-effective intervention method for improving outcomes. A variety of benefits have been found for ONS use, including reduced length of stay [9,10], inpatient episode cost [10], complication rates, [11,12] depressive symptoms [13], and readmission rates [14,15], and improved lean body mass recovery [16]. However, the use of ONS has also been questioned due to low adherence [17,18] and a lack of beneficial results for some patient groups [19]. The effectiveness of nutrition therapy using ONS varies due to unstable patient adherence to the prescription, but a higher adherence to ONS has been associated with a higher energy intake [20,21] and an increase in body weight [21].

ONS often contain macronutrients as energy and protein and micronutrients as vitamins and minerals at varying concentrations. ONS that are registered as pharmaceuticals are only available by prescription, ideally following advice from a doctor. Therefore, individual dietetic assessments take into account a patient's nutritional requirements in order to ensure a tailored prescription. However, some ONS that are registered as foodstuffs are available as over-the-counter purchases in supermarkets or pharmacies without doctors or registered dietitians in Japan. No previous studies have compared the outcomes of two types of ONS "prescription versus over-the-counter ONS".

The aims of this study were the examination of difference between outpatients consumed prescription ONS and those consumed over-the-counter ONS and adherence to prescription ONS prescribed by a doctor and to over-the-counter ONS purchased by themselves.

## Materials and methods

### Study design

Data were obtained from an online cross-sectional survey conducted with patients in Japan. The survey was hosted by the market research company EPOCA Marketing Co., Ltd., which recruited samples from 2.2 million people registered with the company intended to be representative of the Japan population. Prescription ONS were registered in Japan as follows; Elental® (EA Pharma Co., Ltd, Japan), Elental P® (EA Pharma Co., Ltd, Japan), Ensure Liquid® (Abbott Japan Co., Ltd., Japan), Ensure H® (Abbott Japan Co., Ltd.), Enevo® (Abbott Japan Co., Ltd.), Twinline-NF® (Otsuka Pharmaceutical Co., Ltd., Japan), Racol-NF® (Otsuka Pharmaceutical Co., Ltd., Japan). Over-the-counter ONS are registered as foodstuffs ONS in Japan.

The age distribution rate for prescription ONS was examined using the 2nd National Database of Health Insurance Claims and Specific Health Checkups of Japan (NDB) built by the

Ministry of Health, Labor and Welfare of Japan [22]. Given these percentages, questionnaires were collected from the web until the number of patients consuming prescription ONS (including those who also consumed over-the-counter ONS) and those of patients consuming only over-the-counter ONS reached 150 each. The NDB consists mainly of health insurance claims, including basic patient information, such as sex and age, plus items such as the number of insurance points, the name of the illness or injury, medical practice information and drug administration and prescription information.

Patients who consumed prescription ONS combined with over-the-counter ONS and naso-gastric tube or gastrostomy were then excluded. Ultimately, the patients who consumed prescription ONS and those who consumed over-the-counter ONS were defined as the prescription ONS group and over-the-counter ONS group, respectively.

## Ethical approval and studies and informed consent

Respondents had to fill out their name in the questionnaire or had to be identifiable in order to be included in this study and prevent data duplication. Respondents' confidentiality was guaranteed, and privacy policy statements were included in the introduction section of the questionnaire. The study protocol was approved by the Kurume University Ethics Committee (No. 18098).

## Data collection

The questionnaire was started when respondents answered "Yes" to the two following questions: "Have you (or a person in your care) visited a hospital for some illness within the past year?" and "Do you (or a person in your care) currently consume ONS as a hospital outpatient?" The questionnaire then inquired as to whether the respondent was a patient themselves or a caregiver. If the respondent was a caregiver, it asked what their relationship was with the patient. Finally, the survey instrument contained questions about a variety of patient demographic characteristics, such as the age, gender, body mass index (BMI), region, employment and household income (JPY per month). The second part consisted of 18 questions regarding the management and adherence to taking ONS (Table 1).

If respondents answered "Yes" to Q5 "Were you given advice by doctors or registered dietitians?", the respondents went on to answer Q6-Q8: "How often were you advised to take ONS? (e.g. daily, once a week, etc.)", "How many calories of ONS were you advised to consume daily?" and "How long were you advised to take ONS?"

If respondents did not answer "Not determined" or "Unknown" for Q6-Q8, the respondents were then asked Q12 "Do you follow the medical advice (number of times, amount, duration)?" Q12 was then analyzed for each of the different dimensions related to following advice concerning the number of times, amount and duration using a Likert-type 4-point scale: not at all well, slightly well, well and very well. Responses to Q15 "Are you satisfied with your ONS (overall, nutrition, ease of consumption, taste, price, ease of prescription/purchase)?" were analyzed for each of the different dimensions related to satisfaction regarding overall satisfaction, nutrition, ease of drinking, taste, price, ease of obtaining using a Likert-type 7-point scale: very dissatisfied, dissatisfied, slightly dissatisfied, neither satisfied nor dissatisfied, slightly satisfied, satisfied, very satisfied and unknown.

Comparison analyses of the surveys were performed between the prescription and over-the-counter ONS groups. Responses to Q9 and Q11 were also compared between the two groups among patients who were given advice by doctors or registered dietitians. Response to Q10 were compared between the two groups after excluding patients who did not know the amount or type of ONS they consumed.

**Table 1. Survey questions (excluding demographic questions).**

|  | Survey questions |
|---|---|
| Q1 | What is the medical reason for taking the ONS? |
| Q2 | What is the ONS type and brand? |
| Q3 | How much does the ONS cost (JPY per month)? |
| Q4 | Who recommended that the ONS be taken? |
| Q5 | Were you given advice by doctors or registered dietitians? |
| Q6 | How often were you advised to take ONS? (e.g. daily, once a week, etc.) |
| Q7 | How many calories of ONS were you advised to consume daily? |
| Q8 | How long were you advised to take ONS? |
| Q9 | How often do you actually take ONS? (e.g. daily, once a week, etc.) |
| Q10 | How many calories of ONS do you consume daily? |
| Q11 | How long have you been taking ONS? |
| Q12 | Do you follow the medical advice (timing, amount, duration)? |
| Q13 | If not, why do you not follow the medical advice? |
| Q14 | Did you provide support to make it possible to continue taking ONS? |
| Q15 | Are you satisfied with your ONS (overall, nutrition, ease of consumption, taste, price, ease of prescription/purchase)? |
| Q16 | Are you aware of other ONS? |
| Q17 | If Q16"yes", have you ever been recommended to take another ONS? |
| Q18 | If Q16"yes", why do you take the prescription (or over-the-counter) ONS (free comment)? |

## Statistical analyses

Continuous data were presented as mean ± standard deviation, and categorical data were expressed as the number (%). Group differences were tested using the chi-squared test and the Mann-Whitney U test. All of the statistical analyses were performed using the JMP software package (SAS, Cary, NC, USA), and p values of < 0.05 were considered statistically significant.

## Results

### Demographic characteristics

A total of 14.0% of patients prescribed ONS were ≤50 years of age, 22.0% were 51–74 years of age, and 64.0% were ≥75 years of age in the NDB data. Given these percentages, patients were collected. Responses were collected from August 31 to September 18, 2018 until the number of patients consuming prescription ONS (including those who also consumed over-the-counter ONS) and those of patients consuming only over-the-counter ONS reached 150 each. Fig 1 shows the respondents flow of this study, with 83925 potential respondents sent a question-naire. As 82456 respondents did not consume ONS, those were excluded. Three hundred and eighty-seven of these patients had consumed ONS within the past year. Eighty-seven patients had insufficient data. Patients who consume prescription ONS and patients consuming only over-the-counter ONS reached 150 each. Of the patients who consumed prescription ONS, 43 patients who combined with over-the-counter ONS were excluded. Of the patients who consumed over-the-counter ONS, 2 patients who used a nasogastric tube were excluded. Ultimately, 107 patients who matched the inclusion criteria for the prescription ONS group and 148 who matched the criteria for the over-the-counter ONS group were further analyzed (Fig 1).

Table 2 reveals the respondents of the questionnaire. In the prescription ONS group, 36 respondents (33.6%) were patients themselves, and 71 (66.4%) were caregivers, and in the

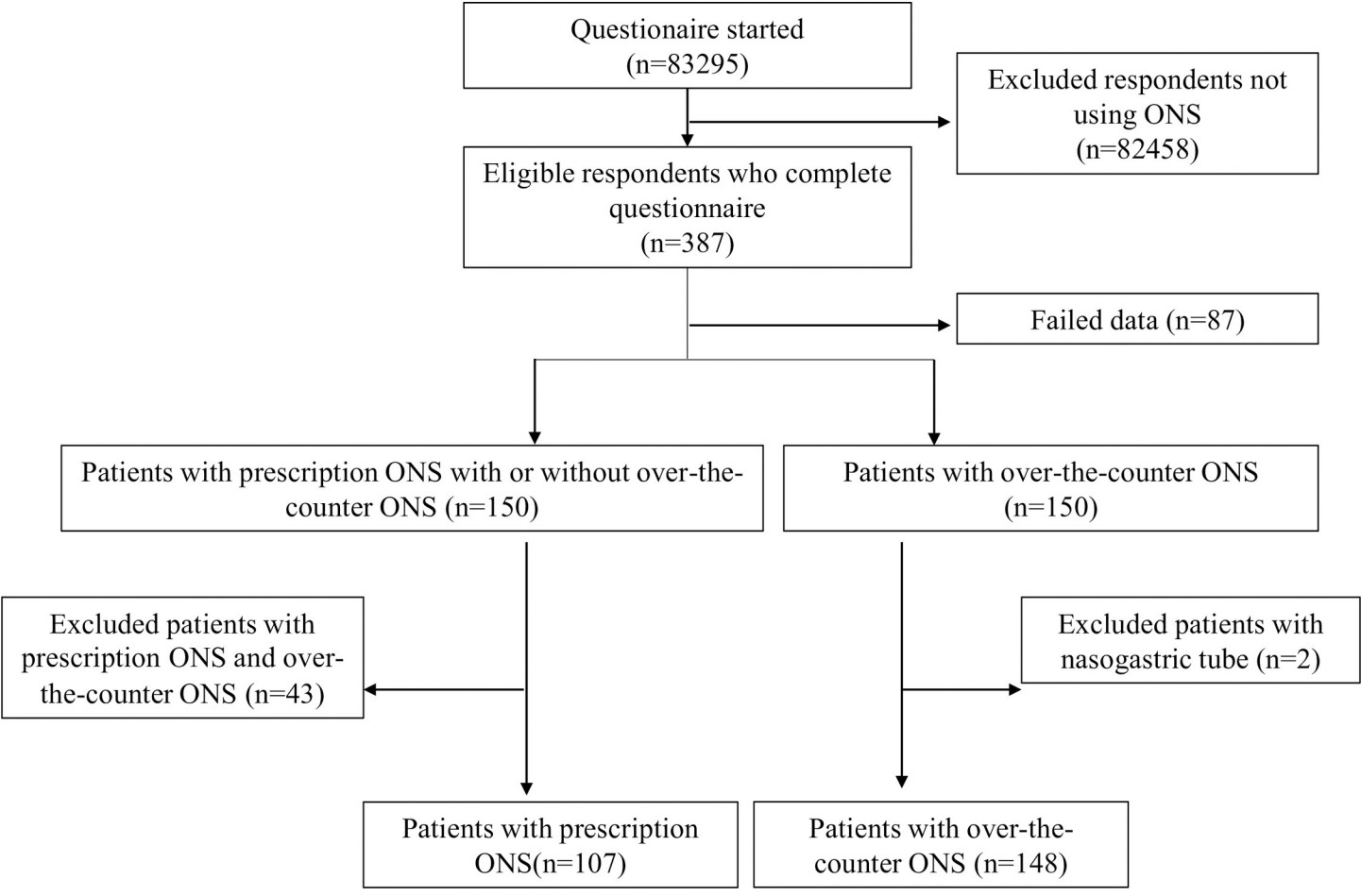

**Fig 1. Flow chart of participants in the overall survey.**

over-the-counter ONS group, 53 respondents (35.8%) were patients themselves, and 95 (64.2%) were caregivers. Caregivers were almost always the parent of the patient (Table 2).

**Table 2. Respondent characteristics.**

|  | Prescription ONS | Over-the-counter ONS | p value |
|---|---|---|---|
| Respondents, n (%) [a] |  |  |  |
| Patients | 36 (33.6) | 53 (35.8) | 0.790 |
| Caregivers | 71 (66.4) | 95 (64.2) |  |
| Family position of patients for caregiver, n (%)[b] |  |  |  |
| Mother or father | 67 (94.4) | 87(91.6) | 0.656 |
| Grandmother or grandfather | 0 (0.0) | 1 (1.1) |  |
| Partner | 4 (5.6) | 4 (4.2) |  |
| Son or daughter | 0 (0.0) | 0 (0.0) |  |
| Brother or sister | 0 (0.0) | 1 (1.1) |  |
| Uncle or aunt | 0 (0.0) | 1 (1.1) |  |

[a]Prescription ONS (n = 107), over-the-counter ONS (n = 148)
[b]Prescription ONS (n = 71), over-the-counter ONS (n = 95)

There was no marked difference in respondents between the prescription and over-the-counter ONS groups.

Table 3 reveals the demographic characteristics of the patients analyzed. There were no significant differences in the age, sex. The BMI of the prescription ONS group was significantly higher than that of the over-the-counter ONS group ($21.1\pm4.38$ kg/m$^2$ vs. $19.9\pm3.75$ kg/m$^2$, $p = 0.016$). Thirty-five patients in the prescription ONS group (32.7%) and 61 in the over-the-counter ONS group (41.2%) had a BMI <18.5 kg/m$^2$.

Table 4 reveals region, employment and household income between the groups. There were no significant differences in employment and household income. However, there were significant differences in the region between the two groups. ($p = 0.042$)

## Medical reasons for taking ONS (review Q1)

In the prescription and over-the-counter ONS groups, the main medical reason for ONS consumption were "malnutrition" (48 patients [44.9%] vs. 63 patients [42.6%] p = 0.798], "frailty" (29 patients [27.1%] vs. 36 patients [24.3%] p = 0.663) and "aging" (25 patients [23.4%] vs. 30 patients [20.3%] p = 0.644). The statistically significant reasons for ONS consumption were "liver disease" (only in the prescription ONS group, 9 patients [8.4%] p <0.0001), "inflammatory bowel disease" (only in the prescription ONS group, 5 patients [4.7%] p = 0.012). The proportion of "No particular disease" for prescription ONS consumption was significantly lower than that for over-the-counter ONS (6 patients [5.6%] vs. 24 patients [16.2%] p = 0.001) (Table 5).

**Type of ONS (review Q2).** Fig 2 shows the types of prescription and over-the-counter ONS. The types of prescription ONS were as follows: Ensure Liquid® (Abbott Japan Co., Ltd., Japan) in 32 patients (29.9%), Ensure H® (Abbott Japan Co., Ltd.) in 23 patients (21.5%), Enevo® (Abbott Japan Co., Ltd.) in 12 patients (11.2%), Racol®-NF (Otsuka Pharmaceutical Co., Ltd., Japan) in 11 patients (10.3%) and unknown in 22 patients (20.6%) (Fig 2A). The majority of the over-the-counter ONS (124 patients [83.8%]) were of the Meibalance® series (Meiji Holdings Co., Ltd.,Japan), and other over-the-counter ONS were taken by under 5% of the over-the-counter ONS group (Fig 2B).

**Table 3. Patients' characteristics.**

| | Prescription ONS (n = 107) | Over-the-counter ONS (n = 148) | p value |
|---|---|---|---|
| Age,n (%) | | | |
| <65 | 18 (16.8) | 32 (21.6) | 0.666 |
| 65–75 | 19 (17.8) | 21 (14.2) | |
| >75 | 70 (65.4) | 95 (64.2) | |
| Sex,n (%) | | | |
| Male | 54 (50.5) | 59 (39.9) | 0.098 |
| Female | 53 (49.5) | 89 (60.1) | |
| Body mass index,mean±SD (kg/m2) | 21.1±4.38 | 19.9±3.75 | 0.016 |
| ≤18.5 kg/m2,n (%) | 35 (32.7) | 61 (41.2) | 0.120 |
| 18.5–25 kg/m2,n (%) | 57 (53.3) | 72 (48.6) | |
| 25–30 kg/m2,n (%) | 9 (8.4) | 13 (8.8) | |
| 30–35 kg/m2,n (%) | 5 (4.7) | 2 (1.4) | |
| 35 kg/m2,n (%) | 1 (0.9) | 0 (0.0) | |

**Table 4. Region,employment and household income of patients.**

| | Prescription ONS (n = 107) | Over-the-counter ONS (n = 148) | p value |
|---|---|---|---|
| Region,n (%) | | | |
| Hokkaido | 3 (2.8) | 6 (4.1) | 0.042 |
| Tohoku | 3 (2.8) | 8 (5.4) | |
| Kanto | 42 (39.3) | 62 (41.9) | |
| Chubu | 23 (21.5) | 24 (16.2) | |
| Kansai | 26 (24.3) | 21 (14.2) | |
| Chugoku | 0 (0) | 10 (6.8) | |
| Shikoku | 3 (2.8) | 2 (1.4) | |
| Kyushu | 7 (6.5) | 15 (10.1) | |
| Employment,n (%) | | | |
| full-time | 38 (35.5) | 47 (31.8) | 0.658 |
| self-employment | 14 (13.1) | 14 (9.5) | |
| part-time | 13 (12.1) | 20 (13.5) | |
| house keeper | 9 (8.4) | 22 (14.9) | |
| umemployment | 12 (11.2) | 19 (12.8) | |
| retirement | 19 (17.8) | 25 (16.9) | |
| other | 2 (1.9) | 1 (0.7) | |
| Household income (JPY),n (%) | | | |
| <3,000,000 | 29 (27.1) | 39 (26.4) | 0.590 |
| 3,000,000–5,000,000 | 26 (24.3) | 47 (31.8) | |
| 5,000,000–7,000,000 | 19 (17.8) | 20 (13.5) | |
| 7,000,000–10,000,000 | 16 (15.0) | 21 (14.2) | |
| 10,000,000–15,000,000 | 9 (8.4) | 12 (8.1) | |
| >15,000,000 | 8 (7.5) | 9 (6.1) | |

## Cost of ONS (review Q3)

The total monthly cost for prescription ONS was 3009±3486 JPY, and that for over-the-counter ONS was 3638±5124 JPY. No significant differences were noted between the groups (p = 0.127) (Table 6).

## Recommendation for ONS (review Q4)

In the prescription ONS group, 72 patients (67.3%) received a recommendation from their prescribing doctors, and 5 patients (4.7%) received a recommendation from other doctors. In the over-the-counter ONS group, 40 patients (27.0%) received a recommendation from a doctor. Recommendations were received from registered dietitians, helpers/care managers/care workers and family more frequently in the over-the-counter ONS group than in the prescription ONS group. Forty-four (29.7%) patients in the over-the-counter ONS group were not recommended from others and 8 (7.5) patients in the prescription ONS group. There were significant differences in the region between the two groups. (p<0.0001) (Table 6).

## Medical advice for ONS (review Q5-8)

In the prescription ONS group, all patients were given medical advice by doctors or registered dietitians, whereas in the prescription ONS group, 46 patients (31.1%) were given advice by doctors or registered dietitians (p<0.001). There were no marked differences between the

**Table 5. Medical reason for taking ONS (Q1).**

| | Prescription ONS (n = 107) | Over-the-counter ONS (n = 148) | p value |
|---|---|---|---|
| Medical reason for taking ONS (Q1), n (%) | | | |
| malnutrition | 48 (44.9) | 63 (42.6) | 0.798 |
| frail | 29 (27.1) | 36 (24.3) | 0.663 |
| cancer (gastroenterogy) | 8 (7.5) | 13 (8.8) | 0.819 |
| cancer (without gastroenterogy) | 3 (2.8) | 11 (7.4) | 0.163 |
| Liver disease | 9 (8.4) | 0 (0.0) | <0.0001 |
| inflammatory bowel disease | 5 (4.7) | 0 (0.0) | 0.012 |
| kidney disease | 3 (2.8) | 4 (2.7) | 1.000 |
| pulmonary disease | 3 (2.8) | 5 (3.4) | 1.000 |
| cardiovascular disease | 4 (7.5) | 13 (8.8) | 0.132 |
| diabetes | 3 (2.8) | 5 (3.4) | 1.000 |
| organic brain disease | 3 (2.8) | 0 (0.0) | 0.073 |
| cerebrovascular disease | 3 (2.8) | 5 (3.4) | 1.000 |
| dimentia | 11 (10.3) | 16 (10.8) | 1.000 |
| psycho-neurologic disease | 4 (7.5) | 3 (2.0) | 0.458 |
| aging | 25 (23.4) | 30 (20.3) | 0.644 |
| others | 10 (9.3) | 11 (7.4) | 0.647 |
| not particular diseae | 6 (5.6) | 24 (16.2) | 0.010 |
| unknown | 8 (7.5) | 5 (3.4) | 0.159 |

groups in the advice about taking ONS, such as the number of times, amount or duration (Table 7).

## Number of times, calories and amount of ONS (review Q9-11)

The prescription ONS group consumed ONS significantly more often than the over-the-counter ONS group (p<0.0001). However, among those given advice by doctors or registered dietitians, there were no significant differences between the groups (p = 0.6253) (Fig 3).

Excluding patients who did not know the amount or type of ONS they consumed, the daily number of ONS calories in the prescription ONS group (n = 74) was significantly higher than in the over-the-counter ONS group (n = 122) (298.02±208.61 vs. 202.62±110.41, p = 0.00044) (Table 8).

In the prescription ONS group, the duration of taking ONS was significantly longer than in the over-the-counter ONS group (p<0.05). However, among those given advice by doctors or registered dietitians, there were no significant differences between the groups (p = 0.812) (Fig 4).

## Following medical advice and support for ONS (review Q12-14)

Regarding adherence to medical advice on taking ONS, no significant differences were seen regarding adherence to consumption frequency in the prescription ONS group (n = 100) and over-the-counter ONS group (n = 43) (Fig 5A), amount in the prescription ONS group (n = 93) and over-the-counter ONS group (n = 41) (Fig 5B) and duration in the prescription ONS group (n = 65) and over-the-counter ONS group (n = 23) (Fig 5C).

Fifty patients in the prescription ONS group responded about why they did not adhere to the medical advice, as follows: doesn't taste good, 12 patients (24.0%); too much to drink, 10 patients (20.0%); difficult to consume, 13 patients (26.0%); side effects, 6 patients (12.0%);

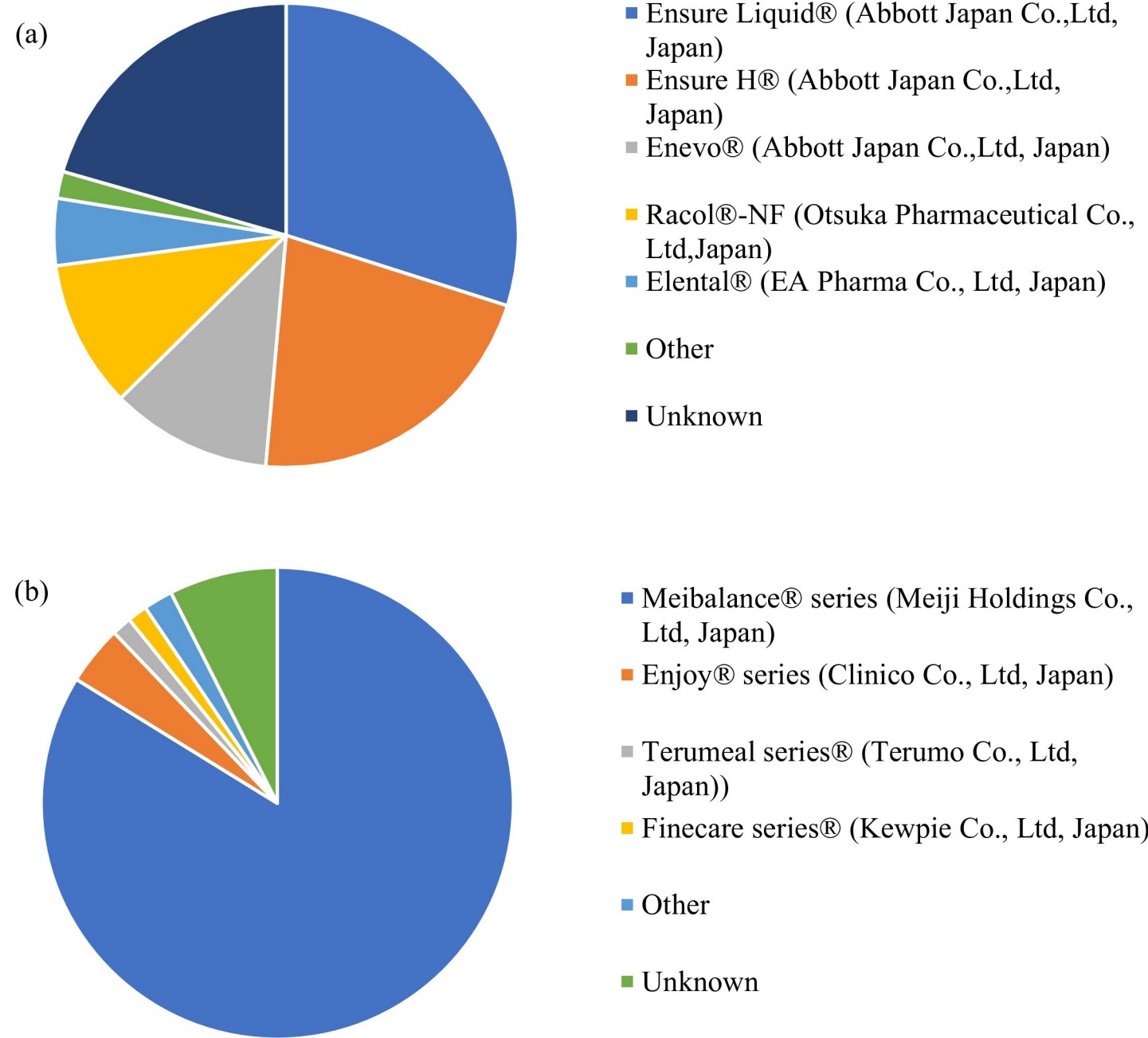

**Fig 2.** ONS types: (a) prescription ONS (n = 107). (b) over-the-counter ONS (n = 148).

don't want to take, 7 patients (14.0%); forget to take, 12 patients (24.0%); not supported by family, 6 patients (12.0%); not supported by medical team, 1 patient (2.0%); and other reasons, 3 patients (6.0%).

Fifteen patients in the over-the-counter ONS group responded about why they did not adhere to the medical advice, as follows: doesn't taste good, 5 patients (33.3%); too much to drink, 2 patients (13.3%); difficult to consume, 3 patients (20.0%); side effects, 2 patients (13.3%); don't want to take, 4 patients (26.7%); forget to take, 4 patients (26.7%); not supported

**Table 6. The result of the survey questions (Q3,4).**

| | Prescription ONS (n = 107) | Over-the-counter ONS (n = 148) | p value |
|---|---|---|---|
| Cost (JPY per month) for ONS (Q3) | | | |
| cost for ONS (JPY/month), mean±SD | 3009±3486 | 3638±5124 | 0.127 |
| <1,000,n (%) | 28 (26.2) | 20 (13.5) | 0.109 |
| 1,000–2,000,n (%) | 21 (19.6) | 30 (20.3) | |
| 2,000–3,000,n (%) | 13 (12.1) | 30 (20.3) | |
| 3,000–4,000,n (%) | 13 (12.1) | 22 (14.9) | |
| 4,000–5,000,n (%) | 3 (2.8) | 4 (2.7) | |
| 5,000–6,000,n (%) | 17 (15.9) | 22 (14.9) | |
| >6,000,n (%) | 12 (11.2) | 20 (13.5) | |
| Recommended to take ONS (Q4), n (%) | | | |
| prescribing doctor | 72 (67.3) | - | - |
| other doctor | 5 (4.7) | - | - |
| doctor | - | 40 (27.0) | - |
| registered dietitian | 2 (1.9) | 21 (14.2) | 0.001 |
| pharmacist | 10 (9.3) | 10 (6.8) | 0.485 |
| Nurse | 10 (9.3) | 20 (13.5) | 0.332 |
| physical therapist / occupational therapist | 3 (2.8) | 4 (2.7) | 1.000 |
| helper/caremanager/care worker | 11 (10.3) | 24 (16.2) | 0.200 |
| others (medical personnel) | 1 (0.9) | 4 (2.7) | 0.402 |
| family | 9 (8.4) | 32 (21.6) | 0.005 |
| friend | 6 (5.6) | 9 (6.1) | 1.000 |
| nobody recommended | 8 (7.5) | 44 (29.7) | <0.0001 |
| other | 2 (1.9) | 0 (0.0) | 0.175 |

by family, 1 patient (6.7%); not supported by medical team, 2 patients (13.3%); and other reasons, 4 patients (26.7%).

The patients in the prescription ONS group who had provided support for continuing taking ONS were significantly more than those of the over-the-counter ONS group who had provided support (48 patients [44.9%] vs. 25 patients [16.9%] p<0.001).

## Satisfaction with ONS (review Q15)

Reports of satisfaction with ease of consumption (Fig 6C) and taste (Fig 6D) were significantly more frequent in the over-the-counter ONS group than in the prescription ONS group (p<0.05 and p<0.05, respectively). However, no significant differences were noted in the overall satisfaction, nutrition, price or availability (Fig 6).

## Awareness of other ONS (review Q16-18)

The awareness of other ONS was significantly different between the groups (p<0.0001). In the prescription ONS group, 59 patients (55.1%) were aware of the existence of over-the-counter ONS, and 38 (35.5%) had consumed over-the-counter ONS. In contrast, in the over-the-counter ONS group, 49 patients (33.1%) knew about prescription ONS, and 14 (9.5%) had consumed prescription ONS. Fewer patients were unaware of the alternative in the over-the-counter ONS group than in the prescription ONS (48 patients [44.9%] vs. 99 patients [66.9%]; Fig 7).

**Table 7. Medical advice for ONS.**

| | Prescription ONS (n = 107) | Over-the-counter ONS (n = 46) | p value |
|---|---|---|---|
| Number of times (Q6) | | | |
| Two or more times a day | 18 (16.8) | 8 (17.4) | 0.663 |
| Once a day | 44 (41.4) | 17 (37.0) | |
| Two to three times a week | 10 (9.3) | 4 (8.7) | |
| Once a week | 1 (0.9) | 3 (6.5) | |
| When you have a low food intake | 27 (25.2) | 11 (23.9) | |
| Not determined | 4 (3.7) | 2 (4.3) | |
| Unknown | 3 (2.8) | 1 (2.2) | |
| Amount (Q7) | | | |
| Indicated capacity | 87 (81.3) | 34 (73.9) | 0.277 |
| As much as possible | 6 (5.6) | 7 (15.2) | |
| Not determined | 6 (5.6) | 2 (4.3) | |
| Unknown | 8 (7.5) | 3 (6.5) | |
| Duration (Q8) | | | |
| ≤2 weeks | 11 (10.3) | 5 (10.9) | 0.218 |
| >2 weeks to 1 month | 8 (7.5) | 4 (8.7) | |
| >1–3 months | 18 (16.8) | 7 (15.2) | |
| >3–6 months | 10 (9.3) | 3 (6.5) | |
| >6–12 months | 6 (5.6) | 0 (0.0) | |
| >12 months | 12 (11.2) | 4 (8.7) | |
| Not determined | 34 (31.8) | 23 (50.0) | |
| Unknown | 8 (7.5) | 0 (0.0) | |

Among the patients who were aware of other ONS (prescription ONS group [n = 59], over-the-counter ONS group [n = 49]), there was no significant difference between the groups in the rate of being recommended or introduced to another ONS (p = 0.091). Thirty-four patients (57.6%) in the prescription ONS group had not been introduced to over-the-counter ONS by their medical team or caregiver, compared with 37 patients (75.5%) in the over-the-counter ONS group (Fig 8).

Among the patients in the prescription ONS group who knew about over-the-counter ONS, responses to "Why do you take your prescription ONS?" were as follows: recommended by a doctor, 14 patients (23.7%); recommended by medical team without doctors, 2 patients (3.4%); medical insurance coverage, 5 patients (8.5%); low price, 4 patients (6.8%); prescription drug, 4 patients (6.8%); easy to drink, 2 patients (3.4%); tasty, 1 patient (1.7%); no reason, 5 patients (8.5%). Among the patients in the over-the-counter ONS group who knew about prescription ONS, responses to "Why do you take your prescription ONS?" were as follows: advised by a doctor, 2 patients (4.1%); low price, 3 patients (6.8%); easy to drink, 5 patients (3.4%); tasty, 1 patient (1.7%); no reason, 5 patients (8.5%).

## Discussion

The prevalence of disease-related malnutrition is reportedly 20%–50% among patients admitted to hospitals [14] and 19% among hospital outpatients [23]. The condition is associated with a decreased quality of life [24,25] and increased length of hospital stay, morbidity, mortality [14,26] and cost of care [26,27].

Baldwin et al. conducted a systematic review and meta-analysis of nutritional intervention with dietary advice and/or oral nutritional supplements during treatment for cancer patients

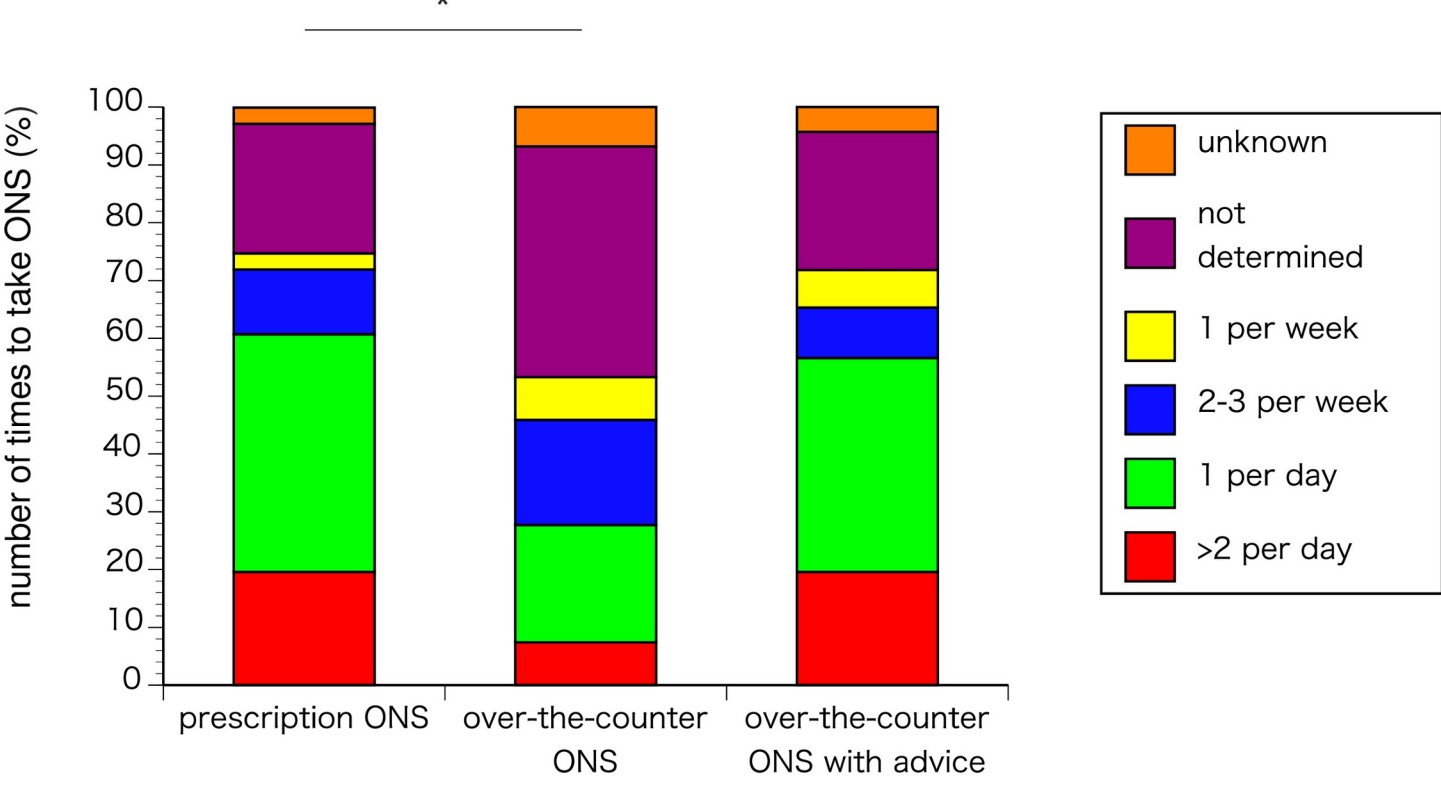

**Fig 3. Number of times to take ONS.** *p<0.0001.

who were malnourished or at nutritional risk. Thirteen studies that included 1414 cancer patients were included in the analysis. Nutritional intervention resulted in statistically significant improvements in weight and energy intake, although no marked differences were observed after removing the studies responsible for heterogeneity. Some aspects of the quality of life, including emotional functioning, dyspnea, loss of appetite, and global quality of life, were improved. Nutritional intervention had no effect on mortality [28]. Concerning advancing age, undernutrition and chronic diseases, Gariballa et al. reported a randomized, double-blind, placebo-controlled trial of ONS. ONS of acutely ill patients improved their nutritional status and led to a statistically significant reduction in the number of non-elective readmissions [29].

For enhancing the food intake, dietary modification and food fortification are necessary. When these measures prove to be ineffective, the provision of ONS is indicated. Good adherence to ONS is essential to the success of nutritional therapy. Regarding factors influencing

**Table 8. Energy intake.**

| | Prescription ONS (n = 74) | Over-the-counter ONS (n = 122) | p value |
|---|---|---|---|
| Energy intake (Q10) (kcal/day), mean±SD | 298.0±208.6 | 202.6±110.4 | <0.0001 |
| <100, n (%) | 13 (17.6) | 20 (16.4) | <0.0001 |
| 101–200, n (%) | 15 (20.3) | 94 (77.0) | |
| 201–300, n (%) | 22 (29.7) | 0 (0.0) | |
| 301–400, n (%) | 13 (17.6) | 2 (1.6) | |
| >400, n (%) | 11 (14.9) | 6 (4.9) | |

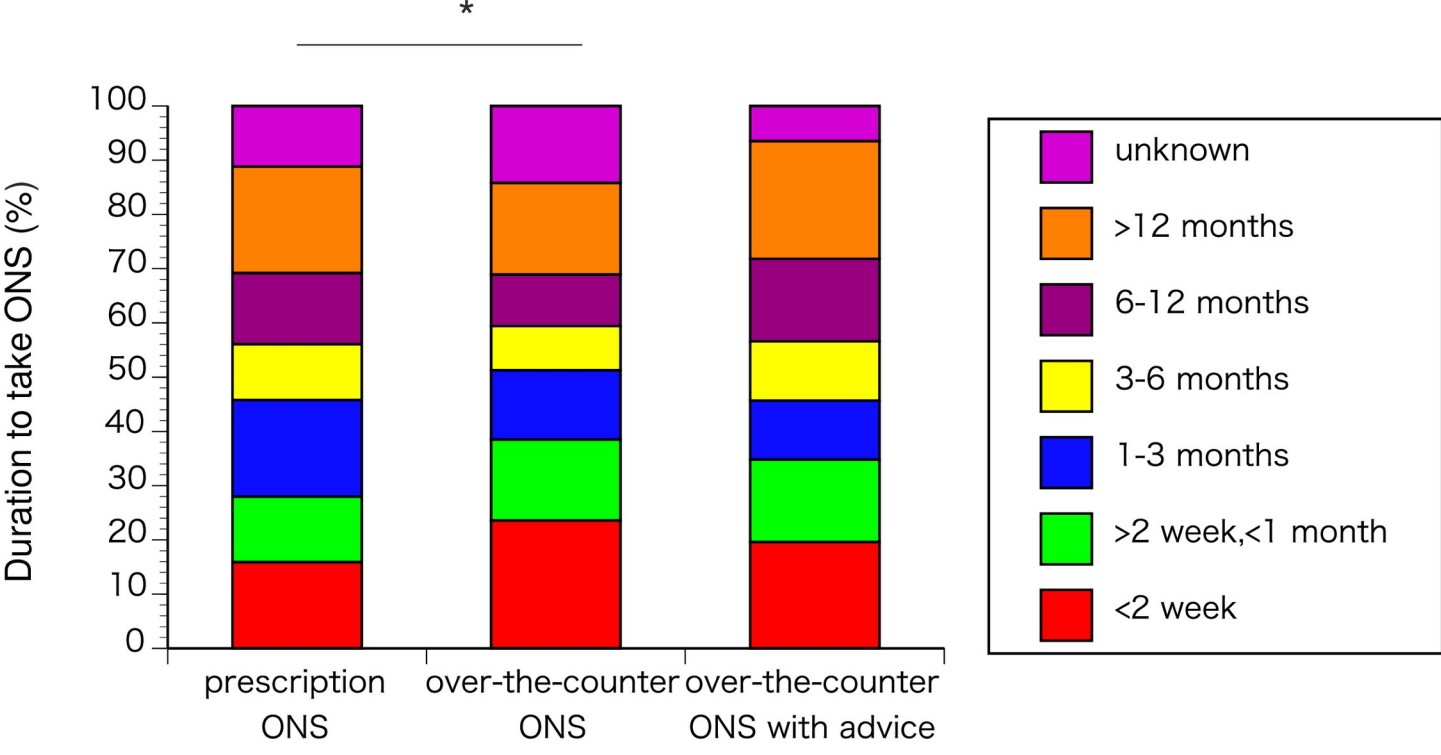

**Fig 4. Duration taking ONS.** *p<0.05.

ONS adherence, a positive association has previously been shown between energy density and ONS adherence [21]. Other influencing factors are suggested to be the duration of ONS usage [30], variety of supplements prescription [21,30], how the supplement is taken [20] and whether or not the patient has been informed of the purpose of the ONS [14].

In the present study, in the prescription ONS group, all patients were given medical advice from doctors or registered dietitians, whereas in the prescription ONS group, only 46 patients (31.1%) were given such advice. Although all patients in the prescription ONS group received a prescription, 67.3% of patients were recommended ONS by their prescribing doctor, and 4.7% were recommended it by another doctor. This seems to suggest that some patients were recommended a prescription ONS by the nutrition support team and others by a multidisciplinary team. Intensive patient education by a nutrition support team to increase the number of feeding opportunities in order to cover the small amount tolerated per occasion as well as for patients to adhere to an adequate ONS program will help reduce body weight loss [31].

In order to achieve compliance, ONS comes in a variety of flavors and textures and can be served at different temperatures, according to patients' tastes, at times they prefer. Furthermore, energy-dense supplements seem to be more easily accepted and effective, as they minimize the volume that must be consumed in order to achieve the desired results [20,32]. Hubbard et al. reported in their systematic review of adherence to ONS that compliance across a heterogeneous group of unmatched studies was positively associated with a greater ONS energy density and total energy intakes, negatively associated with age and unrelated to the amount or duration of ONS prescription [20,32].

Reducing the volume of ONS during medication rounds increased the compliance of patients needing ONS [33]. In the present study, over 80% of over-the-counter ONS were Maibalance®. This formulation is based on the Dietary Reference Intakes for Japanese (2015),

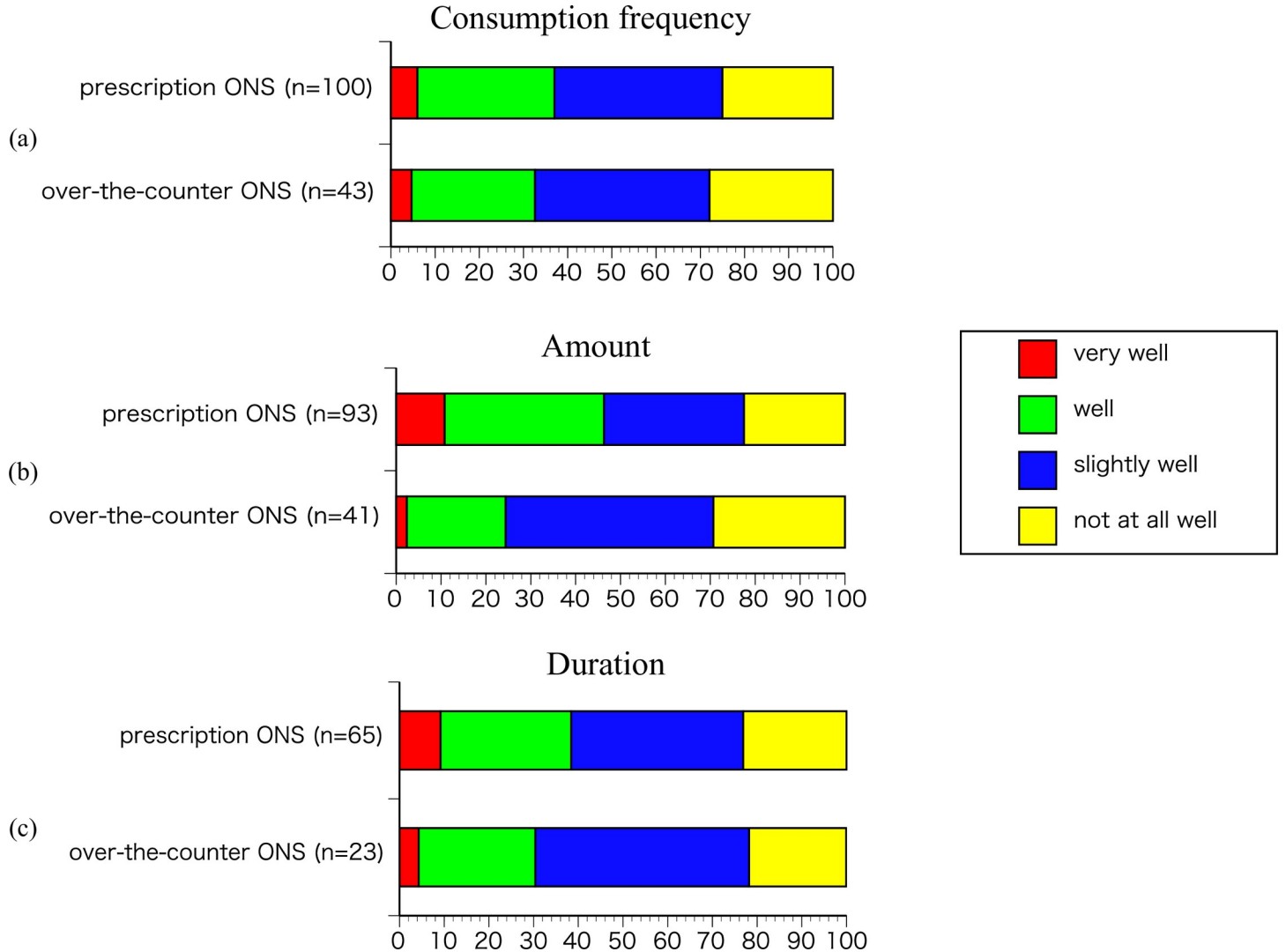

**Fig 5. Adherence to medical advice on taking ONS.** (a); Consumption frequency in the prescription ONS group (n = 100) and over-the-counter ONS group (n = 43). (b) Amount in the prescription ONS group (n = 93) and over-the-counter ONS group (n = 41). (c); Duration in the prescription ONS group (n = 65) and over-the-counter ONS group (n = 23).

comes in several flavors, and can be ingested in small amounts with high calorie counts (200 kcal/125 ml). It is easy to purchase over the counter in supermarkets or pharmacies as well as online. Patients in the over-the-counter ONS group reported significantly higher satisfaction with the ONS ease of consumption and taste than those in the prescription ONS group. However, the number of ONS calories consumed in a day was significantly higher in the prescription ONS group than in the over-the-counter ONS group. It is not possible to consume a sufficient amount of the over-the-counter ONS by taking in patients' own determined, which might be related to the significant difference in the BMI between the two groups.

There has been increasing interest in the effects of ONS on healthcare use and costs. In the acute setting, reductions in the length of hospital stay and complications and a reduction in associated costs have been well documented with ONS [1]. Furthermore, regarding the total healthcare cost and quality of life among patients, ONS use significantly reduces the rate of hospital readmission, especially in older adults [34]. In the present study, while there was no

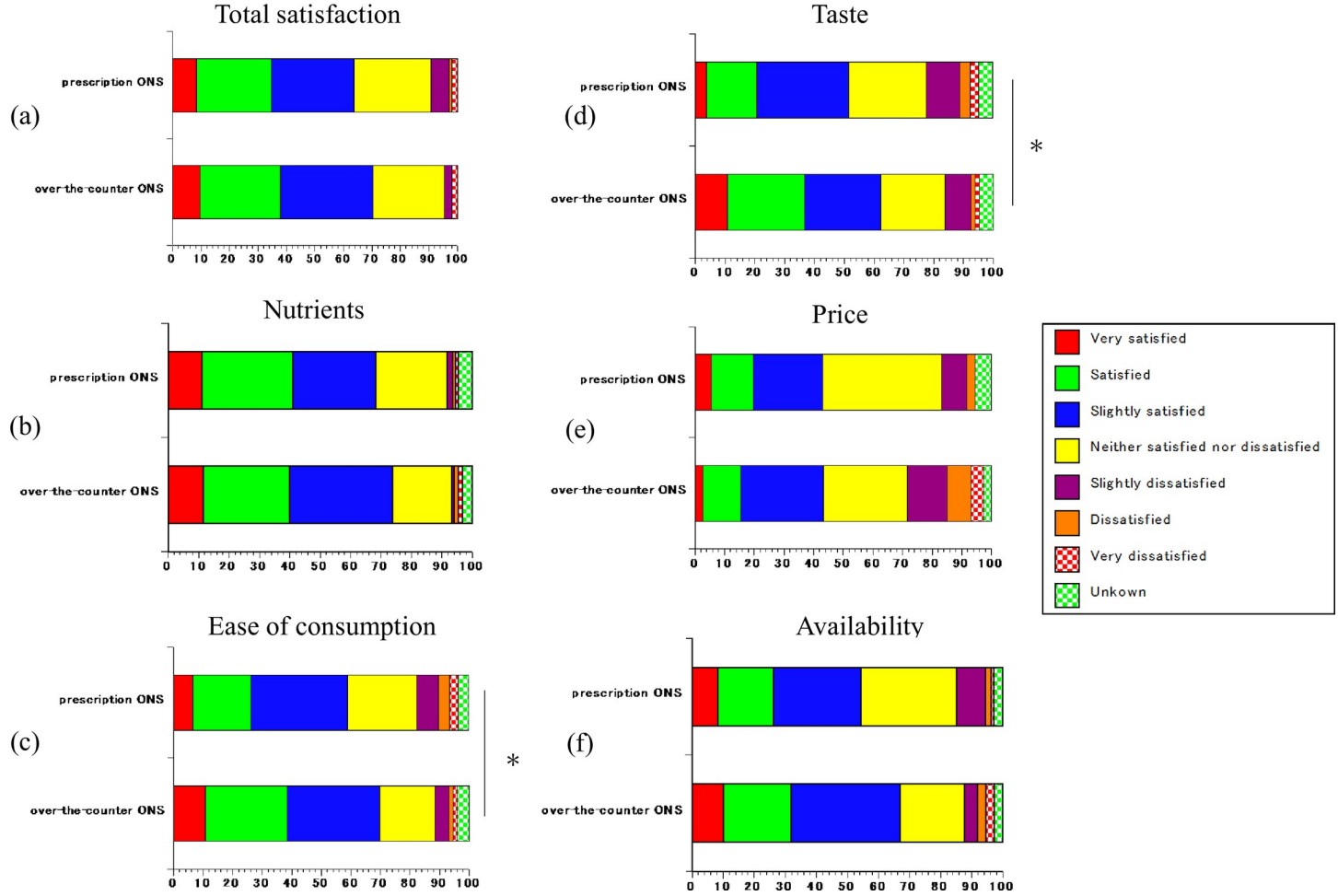

**Fig 6. Satisfaction with ONS.** (a); Overall satisfaction, (b); Nutrition, (c); Ease of consumption, (d); Taste, (e); Price, (f); Availability. *p<0.05.

significant difference between the two groups, the cost in the over-the-counter ONS group was higher than in the prescription ONS group. In Japan, through the national health insurance scheme, patients under 70 years old only have to pay a 30% medical copayment, while those 70–74 years old must pay 20%, and those ≥75 years old must pay 10% (those with income comparable to the current workforce have a copayment of 30%) [35]. If the adherence and calories intake of the two groups were similar, over-the-counter ONS would be much higher than prescription ONS.

In this study, there was a significant difference in the regional spread between the two groups. This may be because doctors and registered dietitians in some regions might recommend one ONS over another, or patients' purchase intentions in certain regions might be higher than in others.

The present study is associated with some limitations due to its wide age distribution and varied health conditions among respondents and small sample size. As this study used an online cross-sectional survey, it was limited to subjects who had internet access. Patients freely consumed both the ONS and ordinary food. Although the data on the ONS intake were accumulated with the utmost care, data on the actual dietary caloric intake were not collected. This study did not categorize ONS into subtypes, such as oligomeric, disease-specific and macronutrient ONS. It might therefore have included patients who did not require ONS treatment,

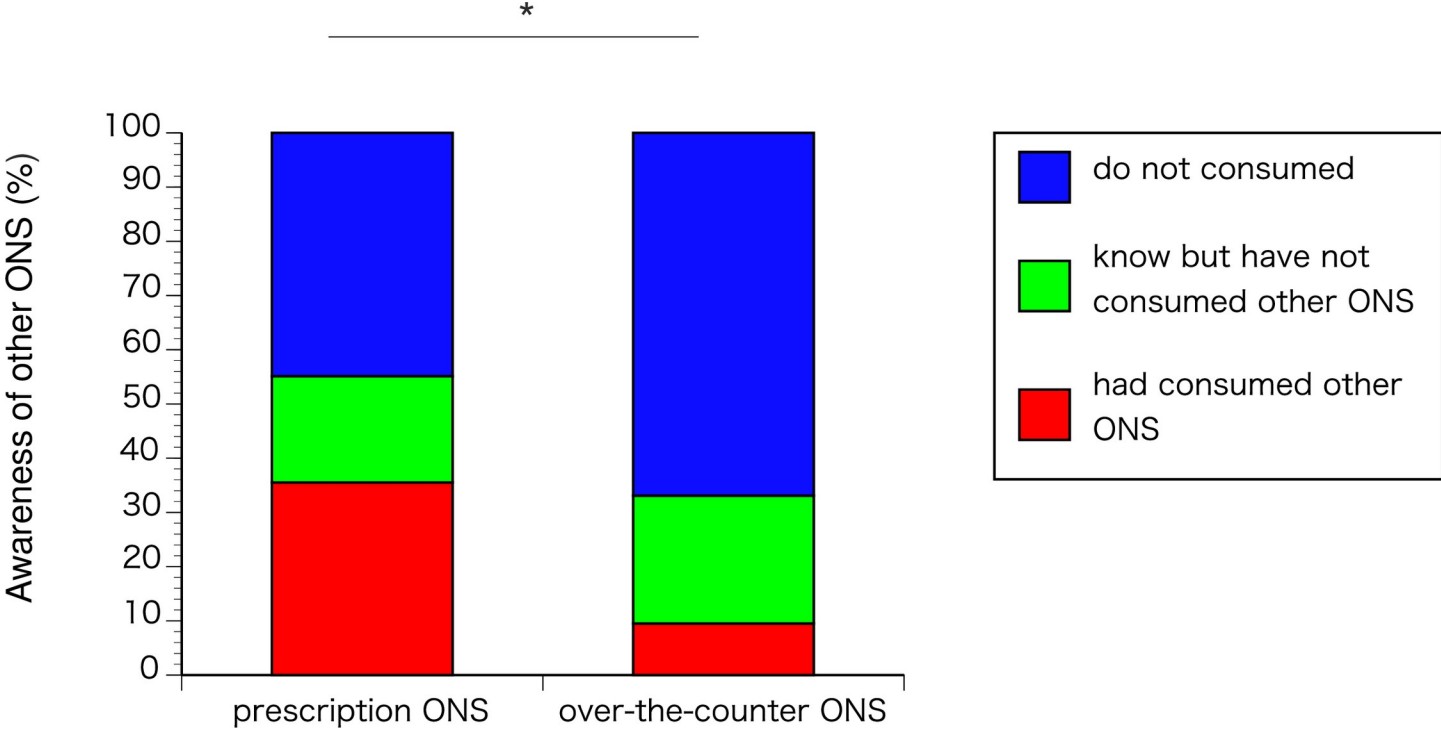

**Fig 7. Awareness of other ONS.** *p<0.0001.

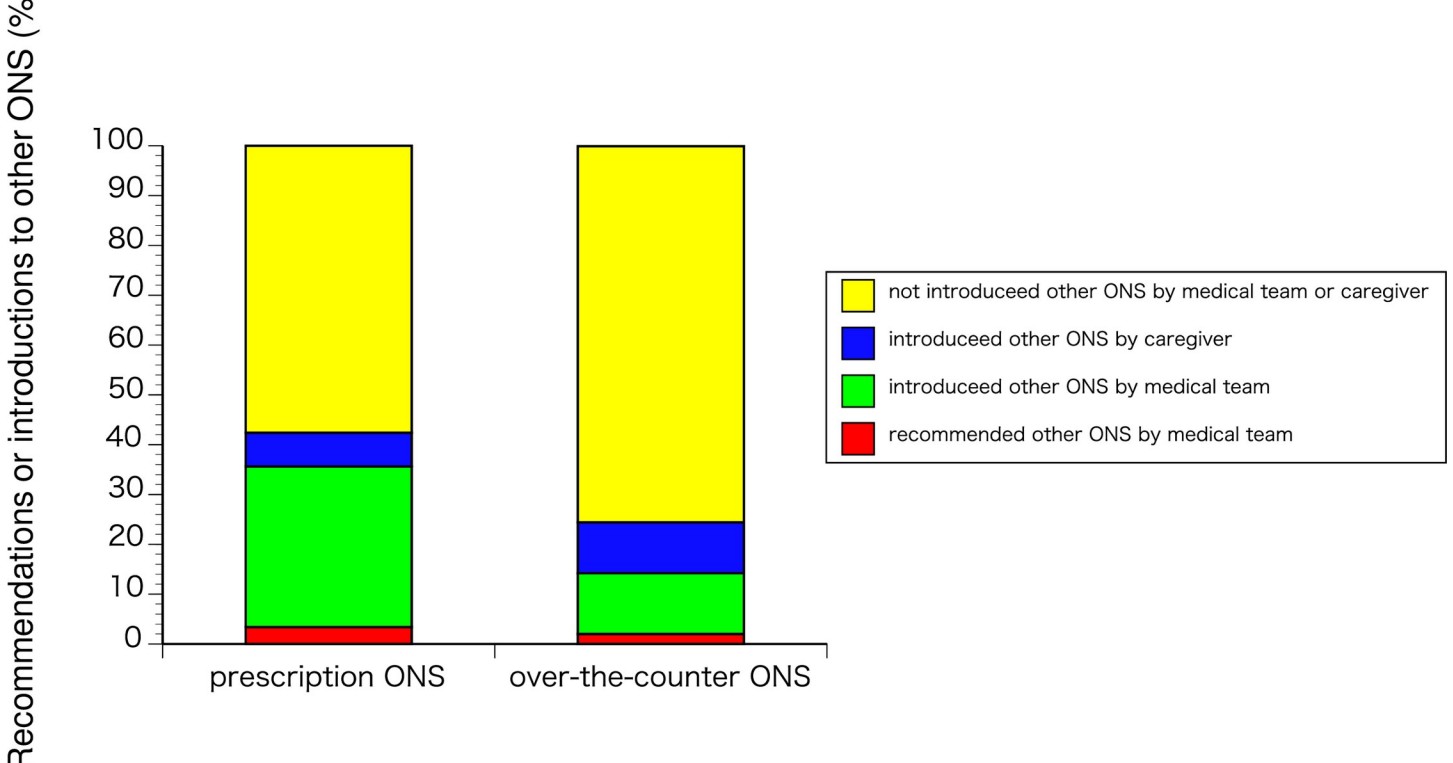

**Fig 8. Recommendations or introductions to other ONS.** Prescription ONS (n = 59) and Over-the-counter ONS (n = 49).

especially the over-the-counter ONS group. It seems also crucial to ensure that patients are offered flavors, textures and/or scents that they like, as these can influence their compliance [36]. Our study did not assess the protein, mineral or vitamin content. In a systematic review and meta-analysis, high-protein supplements were shown to produce clinical benefits, with subsequent economic implications [23]. Furthermore, not only calories but also protein content can significantly influence results.

## Conclusion

Adherence can be improved by encouraging patients and explaining the reasoning and aims of nutritional support. Overall, a greater support by the medical team is still needed in order to maximize adherence to supplementation, especially concerning the calories, timing and period, so that benefits can be achieved and sustained. Consequently, more studies are needed in order to understand the effects of ONSs.

## Author Contributions

**Conceptualization:** Naoki Hashizume.

**Data curation:** Naoki Hashizume.

**Investigation:** Suguru Fukahori, Shinji Ishii, Nobuyuki Saikusa, Yoshinori Koga, Naruki Higashidate.

**Project administration:** Daisuke Masui.

**Visualization:** Saki Sakamoto.

**Writing – original draft:** Naoki Hashizume.

**Writing – review & editing:** Yoshiaki Tanaka, Minoru Yagi.

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
