## [Decision Letter · Decision Letter 0]

31 Jul 2019

PONE-D-19-19397

Adherences to oral nutritional supplementation among hospital outpatients: An online cross-sectional survey in Japan.

PLOS ONE

Dear Dr. Naoki Hashizume,

Thank you for submitting your manuscript to PLOS ONE. After careful consideration, we feel that it has merit but does not fully meet PLOS ONE’s publication criteria as it currently stands. Therefore, we invite you to submit a revised version of the manuscript that addresses the points raised during the review process.

ACADEMIC EDITOR: The reviewers have raised a number of points which we believe major modifications are necessary to improve the manuscript, taking into account the reviewers' remarks.  Please consider and address each of the comments raised by the reviewers before resubmitting the manuscript. This letter should not be construed as implying acceptance, as a revised version will be subject to re-review.

We would appreciate receiving your revised manuscript by Sep 14 2019 11:59PM. To enhance the reproducibility of your results, we recommend that if applicable you deposit your laboratory protocols in protocols.io, where a protocol can be assigned its own identifier (DOI) such that it can be cited independently in the future. For instructions see: http://journals.plos.org/plosone/s/submission-guidelines#loc-laboratory-protocols

We look forward to receiving your revised manuscript.

Kind regards,

Wisit Cheungpasitporn, MD, FACP

University of Mississippi Medical Center

Twitter: @wisit661 Email: wcheungpasitporn@gmail.com 

Academic Editor

PLOS ONE

Journal Requirements:

1. Thank you for including your funding statement; "The funders had no role in study design, data collection and analysis, decision to publish, or preparation of the manuscript."

Please provide an amended Funding Statement that declares *all* the funding or sources of support received during this specific study (whether external or internal to your organization) as detailed online in our guide for authors at http://journals.plos.org/plosone/s/submit-now.  

Please state what role the funders took in the study.  If any authors received a salary from any of your funders, please state which authors and which funder. If the funders had no role, please state: "The funders had no role in study design, data collection and analysis, decision to publish, or preparation of the manuscript."

2. Please amend the manuscript submission data (via Edit Submission) to include author Nobuyuki Saikusa

Reviewers' comments:

Reviewer's Responses to Questions

**Comments to the Author**

1. Is the manuscript technically sound, and do the data support the conclusions?

Reviewer #1: Partly

Reviewer #2: Yes

Reviewer #3: Yes

Reviewer #4: Yes

Reviewer #5: No

2. Has the statistical analysis been performed appropriately and rigorously? 

Reviewer #1: No

Reviewer #2: Yes

Reviewer #3: Yes

Reviewer #4: Yes

Reviewer #5: Yes

3. Have the authors made all data underlying the findings in their manuscript fully available?

Reviewer #1: Yes

Reviewer #2: Yes

Reviewer #3: Yes

Reviewer #4: Yes

Reviewer #5: Yes

4. Is the manuscript presented in an intelligible fashion and written in standard English?

Reviewer #1: Yes

Reviewer #2: Yes

Reviewer #3: No

Reviewer #4: Yes

Reviewer #5: Yes

5. Review Comments to the Author

Reviewer #1: Clearly state the objective of the study in the introduction section.

Add importance of oral nutritional supplements in the introduction section.

Table 3 is too long. If possible, divide into two or more parts.

Table 4 is too long. If possible, divide into two or more parts.

More statistical parameters are applied for the authentication of data.

The references are arranged according to the guidelines of journal.

Reviewer #2: The study design section has some results which can be move to the results section.

Reviewer #3: The manuscript should be revised for linguistic errors.

The oral nutritional supplementation should be specified within the manuscript.

The survey duration is too short and patients number is small.

Reviewer #4: Several studies have found health benefits associated with consumption of oral nutritional supplements (ONS). Hence, their consumption is often recommended, especially in the case of disease-related malnutrition. Hashizume et al. note that the health benefits from ONS consumption vary between patient groups with one of the explanatory factors being the extent to which patients adhere to ONS. Previous studies have found that higher adherence is associated with increase in body weight. With the motivation to find the right strategy to guide patient behavior towards ONS consumption, Hashizume et al. conducted an online survey in Japan. They examined whether the adherence of patients varies according to the kind of ONS that they are consuming — “prescribed” or “purchased”. The “prescribed” ONS are available with prescription only and the “purchased” ONS can be bought over-the-counter. The authors collected data from individuals of different ages, both genders, different BMIs, from different regions of Japan, different occupations and different income levels. They noted that the samples for the two categories of ONS were significantly different in terms of the regional composition of individuals and their BMIs. Although >80,000 individuals started the online questionnaire, the survey effectively included 107 individuals for “prescribed” ONS and 148 for “purchased” ONS. They report their observations on several factors that might distinguish the adherence of those consuming “prescribed” vs “purchased” ONS, e.g., patient characteristics, medical reasons for taking ONS, cost of taking ONS, type of ONS, having received medical advice, number of times and duration over which to take ONS, etc. They also asked the respondents why the patients did not adhere to the medical advice, if they were satisfied with the ONS that they were receiving or if they knew of the alternatives. Based on the responses, the authors conclude that encouraging patients and explaining the reasoning and aims of nutritional support can improve patient adherence. The authors describe the motivation for the survey, its design, the characteristics of data, data analysis methodology, limitations of the survey and provide helpful recommendations to medical teams on how to encourage patient adherence. Overall, the manuscript delivers what it promises in the abstract and the introduction. I have the following questions/suggestions, which I hope the authors will consider:

1. From what I could understand, what the authors call the “prescribed” ONS is also purchased by the patients. Similarly, what they call the “purchased” ONS may also be prescribed/recommended by nutritional experts. I found the terminology confusing. Since the difference between the two categories is that one is available with prescription only while the other is available over-the-counter, I recommend calling the “prescribed” and “purchased” ONS as “prescription” and “over-the-counter” ONS, respectively.

2. On the first page of the abstract, second to the last line, it starts as, “In contrast, in the prescribed ONS group, only 46 patients …”. Did the authors mean purchased ONS group instead?

3. The large majority of individuals who started the questionnaire did not complete it. Is there any systematic reason why this was the case?

4. All the figure legends are repetitions of text also in the main text. I suggest that the figure legends be replaced with a brief title.

Reviewer #5: 1. How did you select participants to invitation to survey

2. Did you have any incentive to complete the survey

3. The main limitation of this study is the low response rate to survey. Therefore, the result cannot represent the nationwide data in Japan

6. PLOS authors have the option to publish the peer review history of their article (what does this mean?). If published, this will include your full peer review and any attached files.

Reviewer #1: Yes: Dr. Ghulam Abbas

Reviewer #2: No

Reviewer #3: No

Reviewer #4: No

Reviewer #5: No

---

## [Author Response · Author response to Decision Letter 0]

17 Aug 2019

Dear reviewers

Thank you for your some comments.We wish to express our appreciation to the reviewers for their insightful comments on our paper. The comments have helped us significantly improve the paper. 

Dear reviewer #1

Q1; Clearly state the objective of the study in the introduction section. 

Add importance of oral nutritional supplements in the introduction section.

A1; We added the section in introduction.

“The aims of this study were the examination of difference between outpatients used prescription ONS and those used over-the-counter ONS and adherence to prescription ONS prescribed by a doctor and to over-the-counter ONS purchased by themselves.”

“A variety of benefits have been found for ONS use, including reduced length of stay, inpatient episode cost, complication rates, depressive symptoms, readmission rates, and improved lean body mass recovery.”

Q2; Table 3 is too long. If possible, divide into two or more parts.

A2; Table3 was divided two parts (Table 3 and Table 4)

Q3; Table 4 is too long. If possible, divide into two or more parts.

A3; Table4 was divided two parts (Table 5 and Table 6)

Q4; More statistical parameters are applied for the authentication of data.

A4; The statistical analysis for 2 groups was used statistically general method according to the question format.

Q5; The references are arranged according to the guidelines of journal.

A5; Absolutely, we check the guidelines of journal.

Reviewer #2: 

Q1; The study design section has some results which can be move to the results section.

A1; We moved to the result about the percentage of patients in NDB and survey duration.

Reviewer #3: 

Q1; The manuscript should be revised for linguistic errors.

A1; This paper was revised by licensed English native speaker.

Q2; The oral nutritional supplementation should be specified within the manuscript.

A2; We added the sentence in method.

“Prescription ONS were registered in Japan as follows; Elental® (EA Pharma Co., Ltd , Japan), Elental P® (EA Pharma Co., Ltd , Japan), Ensure Liquid® (Abbott Japan Co., Ltd., Japan), Ensure H® (Abbott Japan Co., Ltd.), Enevo® (Abbott Japan Co., Ltd.), Twinline-NF® (Otsuka Pharmaceutical Co., Ltd., Japan), Racol-NF® (Otsuka Pharmaceutical Co., Ltd., Japan). Over-the-counter ONS are registered as foodstuffs ONS in Japan.”

Q3; The survey duration is too short and patients number is small.

A3; The number of samples that could be sufficiently quantitatively analyzed was 300 or more in this study. The duration was the period required to have the answer.

Reviewer #4: SI have the following questions/suggestions, which I hope the authors will consider:

Q1; From what I could understand, what the authors call the “prescribed” ONS is also purchased by the patients. Similarly, what they call the “purchased” ONS may also be prescribed/recommended by nutritional experts. I found the terminology confusing. Since the difference between the two categories is that one is available with prescription only while the other is available over-the-counter, I recommend calling the “prescribed” and “purchased” ONS as “prescription” and “over-the-counter” ONS, respectively.

A1; I edited prescribed to prescription, and purchased to over-the-counter

Q2; On the first page of the abstract, second to the last line, it starts as, “In contrast, in the prescribed ONS group, only 46 patients …”. Did the authors mean purchased ONS group instead?

A2; Absolutely, I edited.

Q3; The large majority of individuals who started the questionnaire did not complete it. Is there any systematic reason why this was the case?

A3; Firstly, we collect the patients who visited a hospital for some illness within the past year and currently consume ONS as a hospital outpatient. For collecting 300 patients, we need to start the questionnaire to the large majority of individuals.

Q4. All the figure legends are repetitions of text also in the main text. I suggest that the figure legends be replaced with a brief title.

A4; I edited to a brief title.

Reviewer #5: 

Q1; How did you select participants to invitation to survey?

A1; I edited about the description of selected participants

“The survey was hosted by the market research company EPOCA Marketing Co., Ltd., which recruited samples from 2.2 million people registered with the company intended to be representative of the Japan population.” 

Q2; Did you have any incentive to complete the survey

A2; The aims of this study were the examination of difference between outpatients used prescription ONS and those used over-the-counter ONS and adherence to prescription ONS prescribed by a doctor and to over-the-counter ONS purchased by themselves.

Q3; The main limitation of this study is the low response rate to survey. Therefore, the result cannot represent the nationwide data in Japan

A3; We excluded “nationwide survey”

---

## [Decision Letter · Decision Letter 1]

12 Sep 2019

[EXSCINDED]

Adherences to oral nutritional supplementation among hospital outpatients: An online cross-sectional survey in Japan.

PONE-D-19-19397R1

Dear Dr. Naoki Hashizume,

We are pleased to inform you that your manuscript has been judged scientifically suitable for publication and will be formally accepted for publication once it complies with all outstanding technical requirements.

With kind regards,

Wisit Cheungpasitporn, MD, FACP

University of Mississippi Medical Center

Twitter: @wisit661 Email: wcheungpasitporn@gmail.com 

Academic Editor

PLOS ONE

Additional Editor Comments:

I want to commend the authors on their superb efforts to revise the manuscript according to all reviewers’ suggestions. The quality of the manuscript has improved substantially.

Reviewers' comments:

Reviewer's Responses to Questions

**Comments to the Author**

1. If the authors have adequately addressed your comments raised in a previous round of review and you feel that this manuscript is now acceptable for publication, you may indicate that here to bypass the “Comments to the Author” section, enter your conflict of interest statement in the “Confidential to Editor” section, and submit your "Accept" recommendation.

Reviewer #1: All comments have been addressed

Reviewer #3: All comments have been addressed

Reviewer #4: All comments have been addressed

Reviewer #5: (No Response)

2. Is the manuscript technically sound, and do the data support the conclusions?

Reviewer #1: Yes

Reviewer #3: Yes

Reviewer #4: Yes

Reviewer #5: (No Response)

3. Has the statistical analysis been performed appropriately and rigorously? 

Reviewer #1: Yes

Reviewer #3: Yes

Reviewer #4: Yes

Reviewer #5: (No Response)

4. Have the authors made all data underlying the findings in their manuscript fully available?

Reviewer #1: Yes

Reviewer #3: Yes

Reviewer #4: Yes

Reviewer #5: (No Response)

5. Is the manuscript presented in an intelligible fashion and written in standard English?

Reviewer #1: Yes

Reviewer #3: Yes

Reviewer #4: Yes

Reviewer #5: (No Response)

6. Review Comments to the Author

Reviewer #1: The authors have answered all the queries raised by the reviewer. The article may be accepted in its present form.

Reviewer #3: As the authors addressed the reviewers comments, I suggest acceptance of the manuscript. No further comments are required.

Reviewer #4: (No Response)

Reviewer #5: All of my comments have been addressed as much as possible. I have no further comments to improve this manuscript.

7. PLOS authors have the option to publish the peer review history of their article (what does this mean?). If published, this will include your full peer review and any attached files.

Reviewer #1: Yes: Ghulam Abbas

Reviewer #3: No

Reviewer #4: No

Reviewer #5: No

---

## [Editor Report · Acceptance letter]

19 Sep 2019

PONE-D-19-19397R1 

Adherences to oral nutritional supplementation among hospital outpatients: An online cross-sectional survey in Japan. 

Dear Dr. Hashizume:

I am pleased to inform you that your manuscript has been deemed suitable for publication in PLOS ONE. Congratulations! Your manuscript is now with our production department. 

With kind regards,

on behalf of

Dr. Wisit Cheungpasitporn 

Academic Editor

PLOS ONE